# The Influence of Exogenous CdS Nanoparticles on the Growth and Carbon Assimilation Efficiency of *Escherichia coli*

**DOI:** 10.3390/biology13100847

**Published:** 2024-10-21

**Authors:** Kuo Yang, Yue Yang, Jie Wang, Xiaomeng Huang, Daizong Cui, Min Zhao

**Affiliations:** 1College of Life Science, Northeast Forestry University, Harbin 150040, China; yangkuo9593@163.com (K.Y.); yangyue9593@163.com (Y.Y.); qdndwj@gmail.com (J.W.); a15945096091@163.com (X.H.); 2Key Laboratory for Enzyme and Enzyme-like Material Engineering of Heilongjiang, Harbin 150040, China

**Keywords:** photoelectrons, enhanced electroactivity, metabolic enhancement, oxidative stress, interaction of semiconductors and bacteria

## Abstract

This study investigated the effects of CdS nanoparticles on the growth and me-tabolism of Escherichia coli under different conditions. The results showed that, under illumination, CdS nanoparticles significantly promoted bacterial growth, glucose assimilation, and biomass ac-cumulation, while also enhancing electrochemical properties and upregulating genes related to metabolism and stress response. No significant changes were observed in the dark. The study sug-gests that light-activated CdS nanoparticles can enhance E. coli growth and metabolic efficiency through gene regulation. This provides a new perspective for understanding the interactions be-tween semiconductors and bacteria.

## 1. Introduction

The inhibitory and stimulatory impacts of cadmium sulfide nanoparticles (CdS NPs) on microbial communities are part of a complex interplay that merits detailed examination, especially given its significance in environmental and biotechnological contexts [1,2]. When CdS NPs are photoexcited, they generate electron–hole pairs that have strong antibacterial effects, most of which are caused by the release of heavy metal ions and oxidative stress brought on by photoexcitation [3]. In particular, studies have shown that vertically aligned ZnO@CdS nanorod heterostructures enhance the generation of electron–hole pairs under visible light, leading to the production of reactive oxygen species (ROS). These ROS, such as hydroxyl radicals and superoxide anions, are the primary contributors to oxidative stress and bacterial cell damage. The efficient separation of charges in the ZnO@CdS system allows for increased photocatalytic activity, which significantly enhances the antibacterial effect by damaging cellular structures through oxidative stress [4]. The release of Cd²⁺ ions from these nanoparticles can cause substantial cytotoxicity, mediated not only through direct interactions with cellular proteins, which may lead to the generation of superoxide radicals and consequent cellular damage, but also through impaired protein function and oxidative stress. Interestingly, certain bacterial strains, such as *Lactobacillus acidophilus*, *Bacillus licheniformis*, *Serratia marcescens*, and *Pseudomonas aeruginosa*, have evolved strategies to mitigate this damage by synthesizing cadmium-containing nanoparticles on their surfaces, thus decreasing the toxicity of Cd²⁺ ions [5,6,7,8].

Furthermore, photoexcited electrons and holes from CdS NPs can attack microbial cell membranes and intracellular structures, disrupting essential biomolecules and leading to the accumulation of reactive oxygen species, which cause oxidative stress and structural cellular damage, culminating in cell death [9,10]. Increased ROS levels have been implicated in protein misfolding [11], DNA damage [12], and reduced enzyme activity [13]. Interestingly, research has shown that CdS NPs can induce the upregulation of stress response proteins, such as *EF-Tu* [14], *MutS* [15], and *DnaK* [16], which play crucial roles in protein synthesis, DNA repair, and protein folding, reflecting adaptive cellular mechanisms to deal with Cd²⁺-induced stress and preserve cellular viability [17]. It is important to note that previous studies have also highlighted the potential toxicity of Cd-based nanomaterials in mammals, particularly regarding their negative effects on sperm viability and fertility. Cd-based nanocrystals, such as CdTe, have been shown to cause oxidative stress and DNA damage in sperm cells, leading to reduced fertility rates in animal models [18]. This underscores the potential long-term ecological and biological risks of CdS nanoparticles in broader biological systems.

Conversely, CdS NPs can promote microbial growth under certain conditions. Photoexcited electrons from these nanoparticles enhance intracellular electron transfer and energy flow, augmenting metabolic processes [19]. This has been demonstrated by experiments using natural semiconductor minerals that generate photoelectrons, which have been shown to boost the growth of chemolithoautotrophic and chemoheterotrophic bacteria, such as *Acidithiobacillus ferrooxidans* and *Alcaligenes faecalis* [19,20]. For example, studies have shown that Fe- and Mn-rich oxide coatings on natural minerals can promote microbial growth by driving solar energy conversion into electrons, thereby enhancing microbial metabolic activity through redox reactions [21]. Biohybrid systems that integrate microorganisms and CdS NPs have been shown to have increased metabolic activity under illumination, and the cells can exploit photoexcited electrons to enhance reducing power and support pathways such as the Wood–Ljungdahl pathway for CO_2_ fixation [22]. Additionally, CdS NPs have been shown to significantly enhance the nitrogen fixation ability of *Rhodopseudomonas palustris* through the transfer of photo-induced electrons, resulting in greater biomass accumulation and increased nitrogenase activity [23].

The influence of semiconductor-mediated microbial carbon fixation has presented a significant area of interest in environmental microbiology [24]. This natural semiconductor-mediated effect has the potential to profoundly impact ecological niches by altering microbial community structure and distribution [25,26,27].

In this study, we used *Escherichia coli* as a model organism to investigate the impact of CdS NPs on bacterial growth. The primary objective was to elucidate the dual roles of semiconductors in inducing oxidative stress and promoting metabolic assimilation in biological systems by examining the phototoxicity of CdS NPs and their influence on the enhancement of carbon metabolism cycles. CdS NPs exhibited a dual role, inhibiting microbial growth through toxic effects while enhancing metabolic activities under specific conditions. This balance was crucial for microbial community adaptability and resilience. Understanding these mineral–microbe interactions was essential for comprehending carbon cycling dynamics and microbial ecology in environments enriched with semiconductor nanoparticles.

## 2. Methods

### 2.1. Microbial Strain and Culture Conditions

The test strain was *E. coli* ATCC 15597, purchased from the Global Bioresource Center and cultured in modified M9 medium (1/6 carbon source); the components of the medium are provided below [28] (Table 1):

Bacterial colonies were selected and inoculated into 100 mL of modified M9 medium. Cultures were incubated at 37 °C with shaking at 200 rpm for 12 h. The optical density at 600 nm (OD_600_) was adjusted to 1.0. The resulting bacterial cultures were then used for subsequent experiments.

### 2.2. Synthesis and Characterization of CdS NPs

CdS NPs were synthesized by a modified chemical synthesis method, in which 0.1 mol/L CdCl_2_, Na_2_S and (NaPO_3_)_6_ of solution were prepared and mixed in the ratio of 6:3:20 by volume, and Na_2_S was added slowly to the mixed solution. The mixed solution was sonicated for 30 min by an ultrasonicator (KQ-500DE, Kunshan Ultrasonic instruments Co., Ltd., Kunshan, China), centrifuged at 10,000 rpm for 10 min, and washed with deionized water, and the above steps were repeated three times.

After that, it was washed three times with ethanol, and then the precipitate was dried in a vacuum drying oven after centrifugation, ground into powder, and stored in a dark place.

Images were taken by transmission electron microscopy (TEM) using a JEM-2100 system (JEOL, Tokyo, Japan) which was operated at 200 kV. The sample was adsorbed onto Cu grid and left to dry at room temperature. X-ray diffraction (XRD) was performed using an X-ray diffractometer (X’Pert3 Powder, PANalytical, Almelo, The Netherlands) equipped with Cu Kα irradiation (λ = 1.5406 Å). XRD patterns were recorded from 10 to 90° (2θ) at a scanning speed of 2°/min at 40 kV and 40 mA. UV–Vis DRS was conducted on CdS NPs using a UV–Vis absorption spectrophotometer (TU-1901, Persee, Beijing, China). The integrating sphere was from 200 to 800 nm, and the slit width was 1.0 nm, using BaSO_4_ as a reference. Zeta potential and particle size measurements were performed using a Zetasizer (Nano ZS90, Malvern Instruments, Worcestershire, UK). The FTIR spectrum of the sample was obtained using a Fourier-transform infrared spectrometer (Nicolet iS20, Thermo Fisher Scientific, Waltham, MA, USA).

The photoelectrochemical performance of CdS NPs was measured using an electrochemical workstation (CHI660D, Chenhua Instruments, Shanghai, China) with a standard three-electrode system. A sample-coated glassy carbon electrode was used as the working electrode, an Ag/AgCl electrode as the reference electrode, and a Pt sheet as the counter electrode. Na_2_SO_4_ solution (0.1 mol/L) was used as the electrolyte. The current responses were determined under a 300 W Xe lamp (PLS-SXE300, Perfectlight, Beijing, China) with a 420 nm cutoff filter.

### 2.3. Characterization of E. coli and CdS–E. coli

Synthesized CdS was added to the *E. coli* bacterial culture, with a CdS concentration of 60 mg/L. The initial inoculation density of *E. coli* in all treatment groups was set to OD_600_ = 0.12. The cultures were incubated statically at 28 °C. CdS–*E. coli* and *E. coli* cultures were incubated separately for 48 h. Subsequently, 2 mL of the culture was centrifuged at 14,000× *g* for 10 min. The samples were washed three times with PBS, each wash lasting 20 min. Following the washing steps, 5% glutaraldehyde was added to the samples, which were then agitated using a shaker at 200 rpm for 4 h. The fixed samples were centrifuged again at 14,000× *g* for 10 min and washed three times with PBS. The fixed samples were then dehydrated sequentially using 25%, 50%, 75%, 90%, and 100% ethanol. The dehydrated samples were dispersed onto silicon wafers and sputter-coated with gold to enhance conductivity. The samples were then observed under a JSM-7500F (JEOL, Tokyo, Japan) scanning electron microscope operating at an accelerating voltage of 5.0 kV.

### 2.4. Methodology for Assessing Bacterial Growth in Response to CdS NPs Under Different Conditions

A defined volume of bacterial culture was inoculated into modified M9 medium, thoroughly mixed, and distributed into individual culture flasks. CdS NPs were added at a concentration of 60 mg/L to evaluate their effect on bacterial growth. The initial inoculation density of *E. coli* in all treatment groups was set to OD_600_ = 0.12. The cultures were incubated statically at 28 °C. For the light exposure group, a full-wavelength Xenon lamp was used as the light source, with an intensity of 1500 lux, while the dark group was wrapped in aluminum foil to block light. The experimental setup included the following treatment groups: dark culture (Dark), light culture (Light), dark culture with CdS NPs (Dark-CdS), and light culture with CdS NPs (Light-CdS). Cultures were incubated with shaking and monitored for 12 h. Bacterial growth was assessed by measuring the optical density at 600 nm (OD_600_). Glucose concentration in the culture system was determined using the DNS colorimetric method to monitor the rate of carbon source consumption. Ascorbic acid content in the matrix was measured using the iodometric method to observe the dynamic changes of the sacrificial agent across different treatment groups. The free Cd^2+^ concentration in the CdS–*E. coli* sample was measured using inductively coupled plasma mass spectrometry (ICP-MS) (7800, Agilent Technologies, Santa Clara, CA, USA).

### 2.5. Determination of Key Carbon Cycle Enzyme Activities

After cultivating *E. coli* under optimal growth conditions for 48 h, the samples were centrifuged to obtain the precipitate, which was then resuspended in an extraction buffer. The samples underwent ice-bath ultrasonication (200 W, 3 s sonication intervals with 10 s pauses, repeated 30 times). Subsequently, the samples were centrifuged at 12,000 rpm for 10 min at 4 °C.

The supernatant was collected for further analysis. Enzyme activities of pyruvate kinase (G0811F, Grace Biotechnology Co., Ltd., Suzhou, China), phosphofructokinase (G0808F, Grace Biotechnology Co., Ltd., Suzhou, China), NADP-malate dehydrogenase (G0820F, Grace Biotechnology Co., Ltd., Suzhou, China), and NADP-isocitrate dehydrogenase (G0832F, Grace Biotechnology Co., Ltd., Suzhou, China) in different treatment groups were measured using specific assay kits for each enzyme. All experimental procedures were performed according to the instructions provided with the respective assay kits.

### 2.6. Determination of Antioxidant Enzyme Activities

Cell lysates from the various treatment groups were analyzed to determine antioxidant enzyme activity.

The activities of catalase (CAT), peroxidase (POD), and superoxide dismutase (SOD) during the bacterial growth process were measured using the respective assay kits: catalase assay kit (G0105F, Grace Biotechnology Co., Ltd.), peroxidase assay kit (G0107F, Grace Biotechnology Co., Ltd.), and superoxide dismutase assay kit (G0101F, Grace Biotechnology Co., Ltd.). All experimental procedures were performed according to the instructions provided with the respective assay kits.

### 2.7. Determination of Energy Transfer Intermediates

Cell lysates from the different treatment groups were utilized for the following analyses. The concentrations of key metabolites, including pyruvate (G0807F, Grace Biotechnology Co., Ltd.), ATP (G0815F, Grace Biotechnology Co., Ltd.), and malondialdehyde (G0109F, Grace Biotechnology Co., Ltd.), were measured using specific assay kits. All experimental procedures were performed according to the instructions provided with the respective assay kits.

### 2.8. Quantitative Real-Time Reverse Transcription PCR (qRT-PCR) Analysis

Bacterial cells cultured with and without Cd²⁺ ions were collected under both light and dark conditions at various incubation times. The harvested cells were used to analyze the expression of mRNA for *FtsB*, *FtsZ*, *cydA*, *icdA*, *pykF*, *pfkA*, *grxA*, and *ndh*. The primers used in this study are listed in Appendix A. Total RNA was extracted using a bacterial RNA kit according to the manufacturer’s instructions. The concentration and purity of RNA were determined using a NanoDrop ND-2000 spectrophotometer (NanoDrop Technologies Inc., Wilmington, DE, USA). First-strand cDNA was synthesized using the ReverTra Ace qPCR RT Master Mix with gDNA Remover. Quantitative PCR (qPCR) was performed using the SYBR Green Real-Time PCR Master Mix Kit on a LightCycler 480II Real-Time PCR System (Roche, Basel, Switzerland). The relative expression levels of the target genes were normalized to the expression of the 16S rDNA gene as an internal control. Data were analyzed using the 2^−ΔΔCT^ method. Each sample was tested in triplicate.

### 2.9. Electrochemistry Analysis

All electrochemical analyses were conducted using an electrochemical workstation (CHI 760, Chenhua Instrument, Shanghai, China) and performed in a single-chamber electrochemical reaction cell with a glassy carbon electrode as the working electrode, platinum electrode as the counter electrode, and Ag/AgCl as the reference electrode.

The CV tests on different treatments were performed with the workstation in the range of −1.0 V to 1.0 V at scan rates of 10 mV/s in 0.1 mol L^−1^ Na_2_SO_4_ solution.

ECSA tests of the samples mentioned above were performed based on CV at different scan rates (20, 30, 40, 50, and 60 mV/s) to analyze the electrochemical surface area of the samples.

EIS was performed to determine the resistance of the samples at frequency from 0.01 to 100,000 Hz with Na_2_SO_4_ solution as the electrolyte and a titanium sheet electrode bio-sample as the working electrode.

LSV was operated from −1.0 V to +1.0 V with a scan rate of 10 mV s^−1^ in 0.1 mol L^−1^ Na_2_SO_4_ electrolyte.

### 2.10. Encapsulation and Dye Degradation Experiments

After 48 h of light incubation, 30 mL of the CdS–*E. coli* sample was centrifuged at 14,000× *g* at 4 °C. The sample was washed three times with PBS (pH 7.2) and resuspended in 5 mL of M9 medium. This suspension was mixed with 2 mL of 0.1 g/mL sterilized sodium alginate solution and dropped into 0.1 mol/L CaCl_2_ solution using a syringe to form encapsulated beads. Control groups were prepared using *E. coli* and CdS solutions following the same encapsulation method as the experimental group. Three groups of encapsulated beads (CdS–*E. coli*, CdS only, *E. coli* only) were tested for their ability to degrade 50 mg/L amaranth dye. Decolorization efficiency was measured at a wavelength of 520 nm. After the decolorization experiment, the biological safety of the supernatant was evaluated by adding 100 μL of the decolorized solution to *E. coli* cultures to assess its effects on cell growth and toxicity. Control groups included untreated amaranth dye solution and PBS buffer. Following the initial decolorization experiment, the CdS–*E. coli* beads were washed with PBS and reactivated in M9 medium at 28 °C for 24 h. The dye degradation experiment was repeated to assess the reusability of the encapsulated beads.

## 3. Results

### 3.1. Characterization and Photoelectrochemical Properties of CdS NPs

Appendix A shows a transmission electron microscopy image of the synthesized CdS NPs. This image shows that the sample was constructed from many nanoparticles. In the high-resolution transmission electron microscopy image, we observed lattice fringes, as can be seen in Appendix A. The CdS (111) lattice plane and lattice fringe were observed (inset of Appendix A), and the interfacial spacing was measured to be 0.336 nm [29]. Appendix A shows that CdS NPs have spherical shapes with a size distribution of 80.07 ± 0.38 nm. X-ray diffraction analysis of the CdS crystal structure (Appendix A) revealed diffraction peaks at 26.54°, 43.97°, and 52.16°, corresponding to the (111), (220), and (311) crystal planes, respectively. These peaks matched the standard reference for cubic CdS (JCPDS 10-0454), confirming that the synthesized CdS had a cubic crystalline structure [30].

The zeta potential of the CdS NPs was measured to be −33.93 mV in Appendix A, indicating moderate stability in solution and preventing excessive dispersion. This is crucial for reducing the release of free Cd^2+^ ions. Due to the surface charge of the CdS NPs, they do not readily aggregate or sediment, which helps limit the amount of Cd^2+^ ions released into the culture environment. This, in turn, minimizes the potential toxic effects on bacteria. The FTIR spectrum showed a peak at 515 cm^−1^, indicating the stretching vibration of the Cd–S bond in Appendix A. According to the literature, the stretching vibration of the Cd–S bond typically occurs between 500 and 600 cm^−1^, further confirming the successful synthesis of CdS NPs.

The photoelectrochemical activity of semiconductors is a critical metric for evaluating their performance. Transition photocurrent response testing (Figure 1A) demonstrated that the CdS NPs exhibited a strong response to light and generated a significant photocurrent upon illumination. The photocurrent intensity of CdS NPs remained steady around 1.5 μA/cm^2^, indicating a consistent photoresponse capability. Mott–Schottky analysis (Figure 1B) revealed a flat band potential of −0.74 V (vs. Ag/AgCl). The positive slope of the Mott–Schottky plot confirmed that the CdS NPs were n-type semiconductors. Based on the principle that the conduction band potential of an n-type semiconductor is 0.1 V more positive than its flat band potential, the conduction band potential was calculated to be −0.64 V (vs. Ag/AgCl), and the valence band potential was determined to be 1.83 V (vs. Ag/AgCl). Thus, the synthesized CdS NPs displayed excellent dispersibility, an appropriate band gap, and a rapid photoresponse, indicating their potential for photoelectrochemical applications. In this study, the optical absorption properties of CdS NPs were analyzed using UV–Vis diffuse reflectance spectroscopy (Figure 1C). The absorption edge was observed at approximately 540 nm. From the spectral data, a Tauc plot was derived (Figure 1D), in which the intersection of the linear segment with the X-axis indicated a band gap of 2.47 eV.

Scanning electron microscopy was employed to examine the cultured *E. coli* and CdS–*E. coli* samples in Appendix A. It was observed that *E. coli* alone exhibited normal growth and development, whereas in the CdS–*E. coli* samples, CdS NPs were attached to *E. coli* cell surfaces. This phenomenon indicated that there might have been a potential interaction between CdS NPs and *E. coli*.

The electrochemical impedance spectroscopy (EIS) data for *E. coli* and CdS–*E. coli* under different treatments were fitted using a simulated equivalent circuit to further characterize the electrochemical properties of *E. coli* (Figure 2A). It could be observed that the resistance of *E. coli* decreased 10.55-fold when the culturing system contained CdS NPs, meaning that *E. coli* under the influence of photoelectrons had a higher conductivity. However, in the dark, the different *E. coli* treatment groups did not exhibit any significant changes (Appendix A).

The electrochemical behavior of *E. coli* with or without CdS NPs during different treatments was further assessed under illumination using a series of electrochemical measurements. Cyclic voltammetry analysis showed that under illumination, *E. coli* with CdS NPs had a 33.16% larger capacitance than that of *E. coli* alone (Figure 2B). The higher capacitance of CdS–*E. coli* is indicative of higher electroactivity and the ability to store photoelectrons, which can be used for ATP production in Appendix A. In the dark, the capacitance of *E. coli* was only slightly different when CdS NPs were present, as can be seen in Appendix A. LSV analysis showed that the current intensity of *E. coli* at low potentials increased when the culturing system contained CdS NPs, which might be due to the ability of more photoelectrons released from CdS to flow through the *E. coli* per unit time (Figure 2C,D). However, under dark conditions, the differences in LSV were not significant, as can be seen in Appendix A.

The variation trend of the electrochemical surface area (ECSA) values of *E. coli* and CdS–*E. coli* was consistent with the variation trend of their capacitance values (Appendix A). With CdS NPs under illumination, the ECSA values for *E. coli* increased by 43.69% compared to those of the control group (Appendix A). Having a higher ECSA value means that the CdS–*E. coli* under illumination has more electroactive sites, which gives it a higher electron transfer efficiency. Interestingly, in a dark environment, *E. coli* exhibits higher electroactivity in the absence of CdS NPs (Appendix A).

### 3.2. Growth Characteristics, Antioxidant Enzyme Activity, and Carbon Metabolism in E. coli Under Different Conditions

To evaluate the impact of CdS NPs on the growth of *E. coli*, we monitored bacterial growth under different conditions, as shown in Figure 3. Under illumination, the addition of CdS NPs significantly enhanced the bacterial growth rate, leading to more rapid glucose assimilation from the environment and ultimately resulting in greater biomass accumulation [31]. Conversely, CdS NPs did not improve the bacterial growth rate or carbon consumption rate in the dark, and their effect on final biomass accumulation was not significant. The growth and carbon consumption rates of *E. coli* were comparable under both dark and illuminated conditions, showing no notable differences. Furthermore, the results demonstrated that *E. coli* could not utilize L-ascorbic acid as a growth carbon source. Instead, L-ascorbic acid functioned only as a sacrificial agent.

Under the condition of 60 mg/L CdS NPs, the Cd^2+^ release rate was only 0.71%, with a concentration of 426.78 μg/L. Compared to the minor toxic effects caused by the small amount of Cd^2+^ release, the photoelectrons generated by CdS NPs under illumination had a significantly stronger growth-promoting effect on *E. coli*, as shown in Appendix A. Therefore, the toxic effects of Cd^2+^ were overshadowed by the positive influence of CdS NPs on bacterial growth.

As the concentration of CdS NPs increased, the growth-promoting effect on *E. coli* gradually improved, with the optimal enhancement observed at 60 mg/L of CdS NPs in Appendix A. However, when the concentration increased to 70 mg/L, the growth-promoting effect diminished. This indicated that while CdS NPs significantly promotes bacterial growth at specific concentrations, exceeding a certain threshold leads to a reduction in its positive effect and may even have an inhibitory impact.

These results demonstrate that CdS NPs, when activated by light, facilitate rapid biomass accumulation and faster carbon source assimilation, thereby promoting bacterial growth and metabolism [32,33,34,35].

By examining the effects of exogenous photoelectrons on key enzymes in carbon metabolism, our study substantiated their role in enhancing cellular metabolism. To investigate the effects of photoelectrons on carbon metabolism specifically, we evaluated the activities of pyruvate kinase, phosphofructokinase, NADP-dependent malate dehydrogenase (NADP-MDH), and NADP-dependent isocitrate dehydrogenase (NADP-IDH).

Our results show that there was a significant enhancement in pyruvate kinase activity under illumination when cells were stimulated with CdS NPs, showing a 2.82-fold increase compared to unstimulated cells in the dark. This notable rise in enzymatic activity underlines not just accelerated carbon metabolism but also an increase in enzymatic efficiency, indirectly suggesting an upsurge in ATP synthesis, as seen in Figure 4A.

In contrast, the activity of phosphofructokinase, which is integral to the glycolytic pathway, did not exhibit any change under either lighting condition or with the addition of CdS NPs. This stability implies that although glycolysis and ATP consumption are ongoing, they do not contribute to an increased energy reserve in this scenario, thereby maintaining a balance in the cell’s energetic economy, as seen in Figure 4B.

Additionally, enzyme activity assays revealed that under illumination, the presence of CdS NPs increased the activity of NADP-MDH by 2.11-fold in *E. coli*, whereas NADP-dependent isocitrate dehydrogenase activity dramatically decreased by 9.32-fold in Figure 4C. In the dark, NADP-MDH activity showed no significant variation, but NADP-dependent isocitrate dehydrogenase activity significantly decreased after CdS NPs treatment, as seen in Figure 4D.

These observations suggest that the introduction of exogenous photoelectrons enhances ATP synthesis and elevates reducing power within the cell, as indicated by increased NADP-MDH activity, which facilitates enhanced NADPH synthesis critical for cellular redox balancing. The increase in NADP-IDH activity suggested that the photoelectrons generated by CdS NPs significantly promoted the accumulation of NADPH. This enhancement of the reducing power within *E. coli*, mediated by the photoelectrons from CdS NPs, accelerated the bacterial carbon metabolism rate and promoted cell growth, and through the detection of aldolase, the enhancement of *E. coli* carbon metabolism by photoelectrons was further confirmed.

Collectively, these results underscore that under illumination, the photoelectrons of CdS NPs significantly boost the activities of crucial carbon metabolic enzymes, promoting processes that favor ATP accumulation and enhancing the cellular capacity for exogenous carbon fixation. This enhanced enzyme activity not only underscores the potential metabolic advantages conferred by nanoparticle interaction but also highlights the intricate balance between energy production and carbon assimilation in microbial systems exposed to light [36,37].

Upon the simultaneous generation of large numbers of holes, CdS NPs also produce an equal quantity of photoelectrons. This situation presents an opportunity to explore the direct effects of photoelectrons on microbial cellular functions. The study focused on quantifying the impacts on ATP generation, malondialdehyde (MDA) levels—which correlate with membrane permeability—and pyruvate, a pivotal product in bacterial carbon metabolism.

The experimental results demonstrated that under illumination, *E. coli* cells exposed to CdS NPs displayed a significant enhancement in ATP levels, showing a 1.28-fold increase compared to control *E. coli* cells that were not exposed to NPs, as shown in Figure 5A. This elevation in ATP suggests an increased energy synthesis, facilitated by the influx of exogenous electrons. Notably, no discernible difference in ATP levels was observed in the dark, underscoring the role of light in mediating these effects. Furthermore, the presence of CdS NPs led to a marked increase in MDA content under illumination, indicating enhanced electron transfer processes through cellular membranes, as shown in Figure 5B. This increase not only demonstrates the permeability changes induced by photoelectrons but also suggests the presence of a protective mechanism against oxidative damage from external ROS, highlighted by elevated MDA levels. Higher MDA levels may enhance membrane resistance to lipid peroxidation. Conversely, in the absence of light, the impact of CdS NPs on MDA production was negligible, further emphasizing the photodependent nature of the observed phenomena. Additionally, a substantial rise in pyruvate levels was noted in cells with CdS NPs under illuminated conditions, with a 4.29-fold increase over controls (Figure 5C). This significant enhancement in a key carbon metabolic product suggests that photoelectron entry into the cells markedly improves carbon metabolism transformation and synthesis, thus accelerating biomass accumulation and increasing metabolic efficiency. Collectively, these findings underscore the positive role of exogenous photoelectrons in promoting bacterial energy transport and carbon metabolism and help elucidate the intricate interplay between microbial processes and nanoparticle-induced photoelectronic effects [38].

### 3.3. Assessment of Antioxidant Enzyme Activity and Carbon Metabolic Gene Expression

Upon photoexcitation, semiconductors generate electron–hole pairs, and these holes possess strong oxidative properties and are capable of damaging microorganisms. This study aimed to investigate the response of *E. coli* to photo-oxidative stress induced by CdS NPs by measuring the activities of peroxidase (POD), catalase (CAT), and superoxide dismutase (SOD) under various treatment conditions, as shown in Figure 6. Under illumination, *E. coli* exposed to CdS NPs exhibited a notable increase in the activity of all three antioxidant enzymes, with the most pronounced enhancement observed in CAT activity. Conversely, in the dark, the presence of CdS NPs did not lead to any significant change in antioxidant enzyme activity compared to the control group. These findings indicate that the photoexcitation of CdS NPs results in the generation of holes with strong oxidative potential, which significantly boosts the intrinsic antioxidant enzyme activity in *E. coli*. This increase in enzyme activity is crucial for mitigating the toxic effects of ROS, allowing the microorganisms to minimize oxidative damage, maintain a stable oxidative environment, and support normal growth and metabolism. Overall, our results demonstrated that the strong oxidative stress induced by photoexcited CdS NPs enhanced the antioxidant defense mechanisms in *E. coli*, highlighting the potential implications for microbial resistance to oxidative stress in environments with varying light conditions [39].

Building on previous work, we further evaluated the transcriptional impact of CdS NPs on key growth and metabolic genes in bacterial cells (Figure 7). Specifically, we examined the expression of *FtsN* (regulating Z-ring formation) [40], *pykF* (pyruvate kinase synthesis) [41], and *icdA* (isocitrate dehydrogenase synthesis) [42]. Under illumination with CdS NPs, *FtsN* expression increased by 2.80-fold, *pykF* by 2.90-fold, and *icdA* by 1.37-fold. This significant upregulation indicates that CdS photoelectrons enhance bacterial growth rates and carbon assimilation, accelerating carbon metabolism. Similarly, *ndh* expression, which is essential for ATP synthesis, increased by 1.32-fold [43].

Moreover, the expression of *cydA* (encoding cytochrome bd terminal oxidase) [44] and *grxA* (encoding glutaredoxin) [45] showed notable increases of 1.90-fold and 3.78-fold, respectively, under illumination with CdS NPs. This suggests enhanced electron transfer and oxidative stress resistance in *E. coli*. However, no significant changes in gene expression were observed with CdS NPs in the dark (Appendix A).

These findings demonstrate that light-activated CdS NPs significantly upregulate the transcription of key metabolic and stress response genes in *E. coli*, promoting growth and enhancing oxidative stress management.

### 3.4. Dye Degradation, Reusability, and Biological Safety Assessment

The CdS–*E. coli* encapsulated beads demonstrated the highest decolorization efficiency in the amaranth dye degradation experiment, achieving a rate of 41.06% within 24 h (Appendix A). In contrast, CdS alone achieved a decolorization rate of 14.62%, while *E. coli* alone showed a rate of 17.66%. The amaranth dye itself did not exhibit significant spontaneous decolorization. When the CdS–*E. coli* encapsulated beads were reused across multiple cycles, the decolorization efficiency remained stable at approximately 40% during the first three cycles, indicating strong stability and reusability in treating dye pollutants (Appendix A). Additionally, the biological toxicity assessment of the supernatant after decolorization showed that the treated supernatant did not exhibit significant toxicity, whereas the original amaranth dye solution significantly inhibited *E. coli* growth in Appendix A.

## 4. Discussion

The interaction between CdS NPs and *E. coli*, particularly their influence on growth and carbon metabolism, has profound implications for microbial ecology and biotechnology. Our study highlighted the significant role of photoelectrons in enhancing the electroactivity of *E. coli*. Upon light exposure, CdS NPs generated electron–hole pairs, releasing photoelectrons that significantly increased the electroactivity of *E. coli*. This increased electroactivity facilitated better electron acceptance by the bacteria, leading to more robust growth and enhanced metabolic processes.

Enhanced electroactivity resulted in increased ATP production, which was crucial for various cellular processes, including growth and reproduction [46,47]. The promotion of bacterial growth under illumination indicated that nanoparticles could significantly influence microbial population dynamics over time. Enhanced growth rates led to faster colonization and biofilm formation, important in both natural and engineered environments [48].

The acceleration of key metabolic activities, such as carbon metabolism, was another critical outcome of increased electroactivity. Enhanced electroactivity resulted in more efficient carbon assimilation, leading to rapid biomass accumulation and higher metabolic rates [49]. This improved metabolic rate has several significant implications.

Firstly, in natural environments, such as soil and aquatic ecosystems, the enhanced growth and metabolic efficiency of bacteria could alter microbial community composition and ecological functions. Bacteria with higher electroactivity and faster metabolism might become more dominant, altering ecological niches and roles within the microbial community. This shift could impact nutrient cycling, organic matter decomposition, and overall ecosystem dynamics, potentially leading to changes in soil fertility and ecosystem productivity [50]. The study indicated that CdS NPs, being spherical and negatively charged (−33.93 mV), were less likely to physically damage bacterial membranes or aggregate around *E. coli*, making direct contact or wrapping mechanisms unlikely [51,52]. The primary toxicity mechanism was the increased ROS production, which *E. coli* countered by upregulating antioxidant enzymes. This strong antioxidative response mitigated the potential negative effects of ROS, allowing bacterial growth to continue [53]. Although Cd^2+^ release reached 426.78 μg/L, the toxic impact was outweighed by the significant growth-promoting effects of photoelectrons, especially their enhancement of bacterial glycolysis [54]. Nanobubble generation was irrelevant in our oxygen-limited environment [55].

Secondly, the ability to manipulate bacterial metabolism using CdS NPs has practical applications in biotechnology. Enhanced metabolic rates could be harnessed to increase the production of valuable metabolites in engineered systems. For example, in bioreactors used for biofuel production, the accelerated carbon metabolism of bacteria could lead to higher yields and more efficient biofuel generation. This would make biofuel production more sustainable and economically viable, contributing to renewable energy solutions [56].

Furthermore, the increased electroactivity and metabolic efficiency induced by CdS NPs could be leveraged for environmental remediation. Bacteria with enhanced capabilities could be employed to degrade environmental pollutants more effectively. However, in our experiments, we observed that although the Cd^2+^ release concentration was 426.78 μg/L, which was lower than the safety threshold for kidney toxicity in some mammals and therefore not expected to have significant toxic effects, the photoelectrons generated by CdS NPs under light conditions significantly promoted the growth of *E. coli*, masking the inhibitory effects of Cd^2+^. Applying the CdS–*Shewanella oneidensis* encapsulated system effectively degraded dyes such as trypan blue with a degradation rate of up to 95% and also demonstrated high efficiency with naphthol green B under photocatalytic conditions. The encapsulation prevented CdS NPs leakage, ensuring biological safety, making it a viable and sustainable solution for wastewater treatment [57].

The upregulation of antioxidant enzymes such as CAT, SOD, and POD in response to photoelectrons indicated an adaptive mechanism that not only ensures survival but also improves metabolic efficiency [58]. This resilience has long-term ecological implications. Enhanced antioxidative capabilities make bacteria more competitive in their natural habitats, particularly in polluted environments where oxidative stress is prevalent [59]. This could lead to shifts in microbial community structures, favoring species or strains that effectively manage oxidative stress, thus influencing ecosystem dynamics and functions [60].

In conclusion, the interaction between CdS NPs and *E. coli* not only enhanced bacterial electroactivity and growth but also accelerated key metabolic processes like carbon metabolism. These effects have broad ecological and biotechnological implications, including potential shifts in microbial community structure and improved efficiency in biotechnological applications. Understanding these interactions is crucial for harnessing the full potential of nanotechnology in environmental and industrial contexts [61].

## 5. Conclusions

This study demonstrated that CdS NPs significantly enhance the growth and metabolic activity of *E. coli* under light conditions. The CdS NPs exhibited strong photoelectrochemical properties, which contributed to increased glucose assimilation, biomass accumulation, and electroactivity, particularly in capacitance and electron transfer efficiency. Photoexcited CdS NPs induced higher ATP levels, enhanced antioxidant enzyme activities, and elevated pyruvate production in *E. coli*, highlighting the beneficial impact of photoelectrons on bacterial energy and carbon metabolism. These findings underscore the potential of CdS NPs in applications requiring enhanced microbial growth and metabolic processes facilitated by light-activated nanoparticles.

## Figures and Tables

**Figure 1 biology-13-00847-f001:**
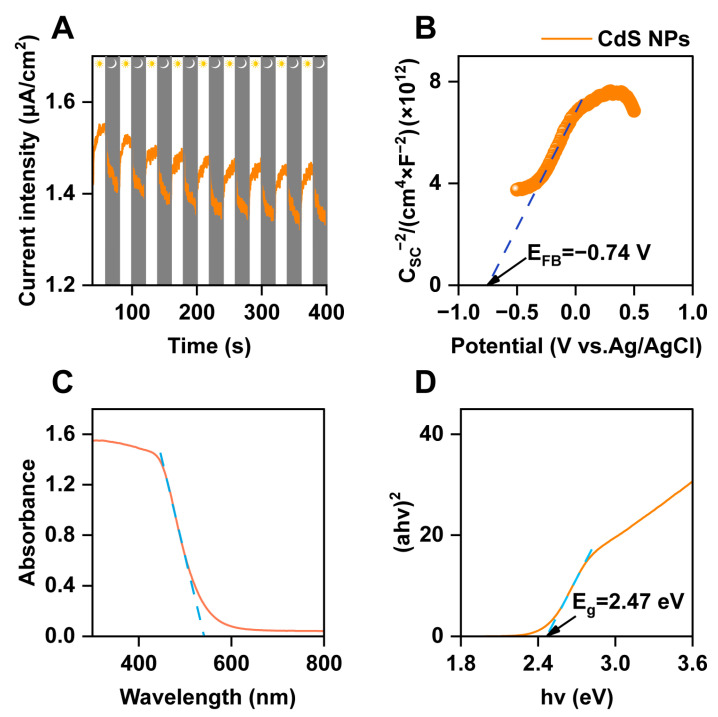
Photoelectrochemical characteristics of CdS NPs. (**A**) Transition photocurrent of CdS NPs under a Xe lamp irradiation; (**B**) the Mott–Schottky curve of CdS NPs; (**C**) UV–Vis diffuse reflection spectra; (**D**) the bandgap value of CdS NPs in Tauc plot.

**Figure 2 biology-13-00847-f002:**
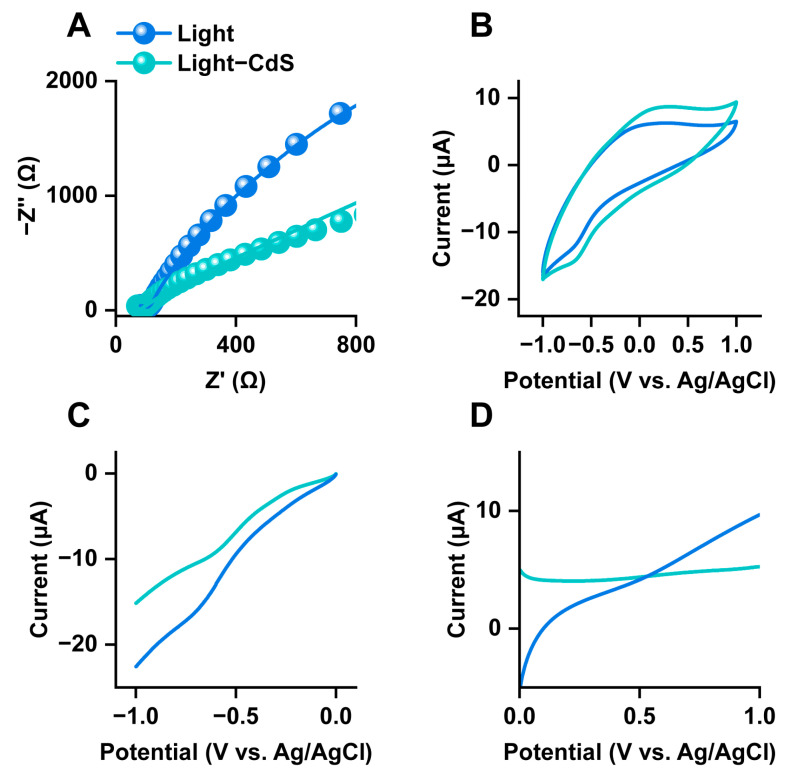
(**A**) Nyquist plot of the samples in light environment and fitted data (solid line). (**B**) Cyclic voltammogram of *E. coli* cultured under light condition with or without CdS NPs. (**C**) Cathodic LSV curves on *E. coli* with or without CdS NPs in light groups. (**D**) Anodic LSV curves on *E. coli* with or without CdS NPs in light groups.

**Figure 3 biology-13-00847-f003:**
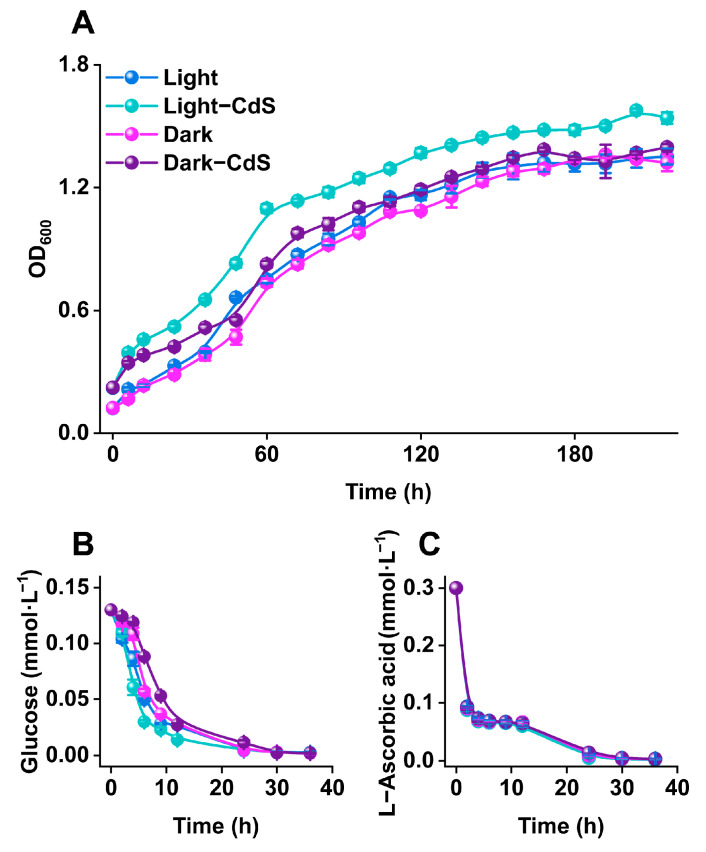
(**A**) The growth of *E. coli* under different culture conditions and the variations in (**B**) glucose and (**C**) L-ascorbic acid concentrations in the culture medium.

**Figure 4 biology-13-00847-f004:**
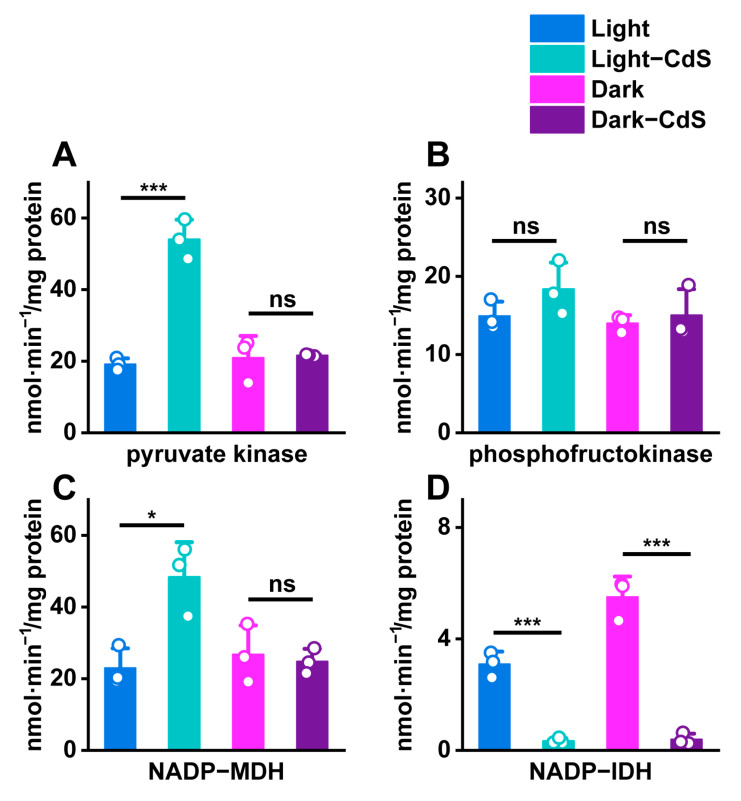
Differences in the activity units of key enzymes in carbon metabolism by different treatment groups: (**A**) pyruvate kinase, (**B**) phosphofructokinase, (**C**) NADP-MDH, and (**D**) NADP-IDH. ns > 0.05, * *p* ≤ 0.05, *** *p* ≤ 0.001, *n* = 3 for each group.

**Figure 5 biology-13-00847-f005:**
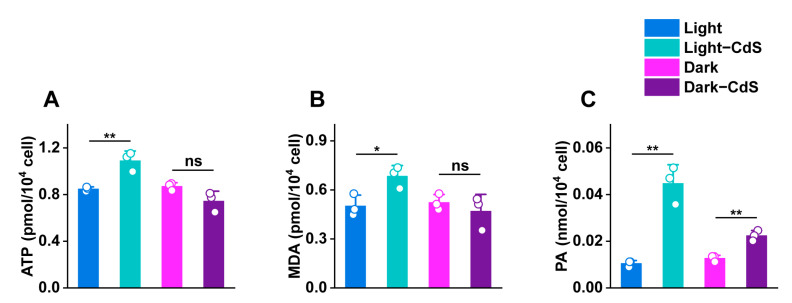
Differences in the concentrations of (**A**) ATP, (**B**) MDA, and (**C**) PA by different treatment groups. ns > 0.05, * *p* ≤ 0.05, ** *p* ≤ 0.01, *n* = 3 for each group.

**Figure 6 biology-13-00847-f006:**
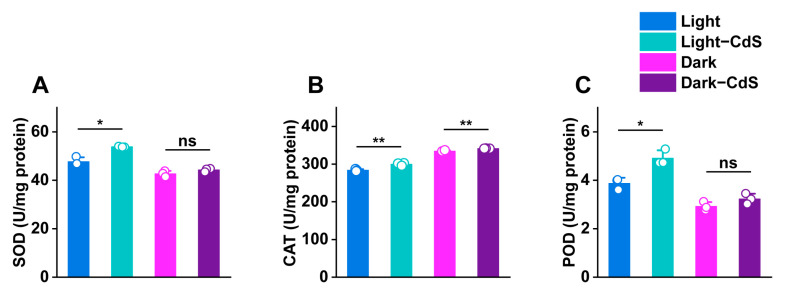
Differences in antioxidant enzyme activities in different treatment groups: (**A**) SOD, (**B**) CAT, and (**C**) POD. ns > 0.05, * *p* ≤ 0.05, ** *p* ≤ 0.01, *n* = 3 for each group.

**Figure 7 biology-13-00847-f007:**
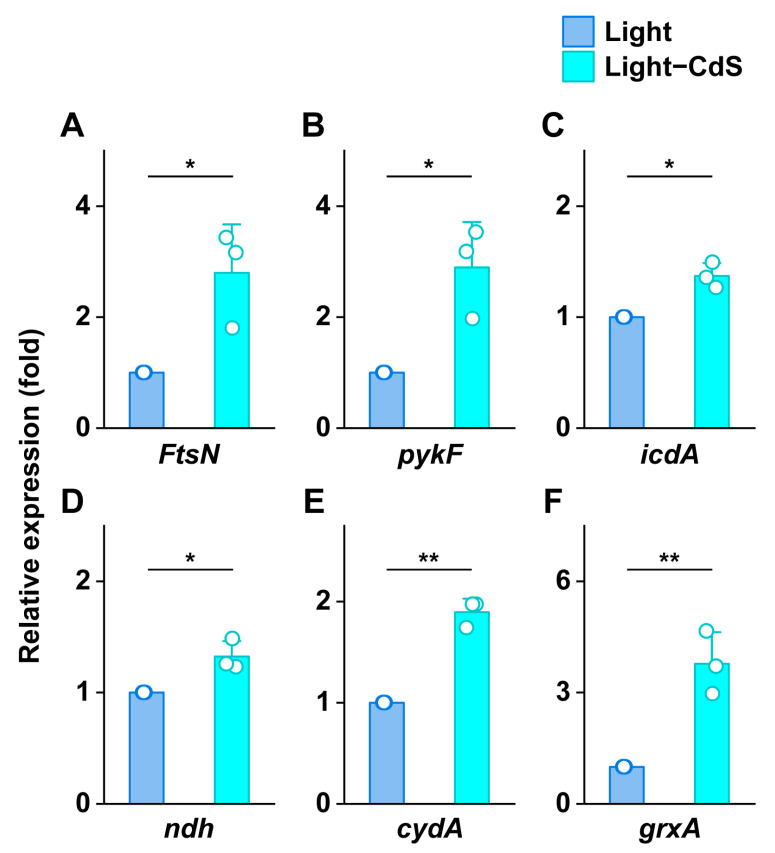
The relative expression levels of (**A**) *FtsZ*, (**B**) *pykF*, (**C**) *icdA*, (**D**) *ndh*, (**E**) *cydA*, and (**F**) *grxA* in *E. coli* under light environment. * *p* ≤ 0.05, ** *p* ≤ 0.01, *n* = 3 for each group.

**Table 1 biology-13-00847-t001:** Modified M9 medium.

Components	mL L^−1^
MgSO_4_·7H_2_O solution *	2 mL
CaCl_2_ solution **	100 μL
5 × M9 salt solution ***	200 mL
Glucose solution ****	20 mL
HCl	Adjust pH to 7.0
MgSO_4_·7H_2_O	246.47
CaCl_2_	111
Na_2_HPO_4_·12H_2_O	19.6
KH_2_PO_4_	3
NaCl	0.5
NH_4_Cl	1
Glucose	33.3

* From filter-sterilized stock solution containing (g L^−1^). ** From filter-sterilized stock solution containing (g L^−1^). *** From filter-sterilized stock solution containing (g L^−1^). **** From filter-sterilized stock solution containing (g L^−1^).

## Data Availability

Data are contained within the article and Appendix A.

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
