# Peer review of "The Influence of Exogenous CdS Nanoparticles on the Growth and Carbon Assimilation Efficiency of Escherichia coli"

_biology, 2024, doi:10.3390/biology13100847_

Round 1
Reviewer 1 Report
Comments and Suggestions for Authors
The authors demonstrate the growth promotion of E. coli using CdS nanoparticles. The research design is appropriate, and content supports the conclusion; however, I have few suggestions to improve the manuscript text.
1. The introduction should include the practical advantages of bacterial growth promotion. Briefly, it was presented in discussion but lacks practical relevancy. For instance, how will CdS NPs be available in the environment to interact with bacteria?
2. The concentration effect of nanoparticles on bacterial growth is not clear. Maybe, at higher concentrations, an opposite effect will be observed. The author must include the concentration effect.
3. In section 2.3, it is not clear, how CdS-E.Coli was prepared.
4. TEM results show aggregates and make it difficult to understand the size and shape.
5. Authors should elaborate on the practical relevancy of the work with logical explanations. For instance, in discussion, line 416-418, degradation of environmental pollutants with E.Coli? How? You first, treat with NPs, and release the bacteria in the environment? How long does the effect remain in natural conditions? Is E.coli, appropriate bacteria to do this? If not, that choose relevant bacteria for study and than only made such claims.
Comments on the Quality of English LanguageEnglish is readable and understandable
Author Response
Dear editor:
Thank you very much for reviewing my paper and giving me so many useful comments and suggestions. I have already revised my paper accordingly, and the details are as following:
Reviewer 1:
- The introduction should include the practical advantages of bacterial growth promotion. Briefly, it was presented in discussion but lacks practical relevancy. For instance, how will CdS NPs be available in the environment to interact with bacteria?
Response:
Indeed, it is necessary to further elaborate on the interaction between CdS nanoparticles and bacteria. I would like to address this in more detail as follows:
Although there are relatively few studies on the interaction between natural CdS minerals and bacteria, research has shown that certain mineral films, such as Fe- and Mn-rich oxide coatings, can promote bacterial growth and metabolism through their semiconductor properties. For example, the study by Lu et al. demonstrated that these mineral coatings can drive microbial metabolism by converting solar energy into electrons, thereby enhancing microbial growth[1].
In the context of our research, there is already evidence showing that CdS nanoparticles can significantly enhance the growth and metabolic capabilities of certain photosynthetic bacteria through electron transfer. Research conducted by Wong et al. demonstrated that CdS nanoparticles, when exposed to light, can significantly enhance the nitrogen fixation ability of Rhodopseudomonas palustris. This study shows that the CdS hybrid system promotes nitrogen fixation and biomass accumulation by providing additional reducing power to nitrogenase through the transfer of photo-induced electrons[2]. These findings illustrate the potential of CdS nanoparticles to promote bacterial growth by facilitating electron transfer in environmental conditions.
These suggested that there might be previously overlooked connections between microorganisms and natural semiconductor minerals.
We have incorporated these details into the manuscript, specifically at line 59-65 and 72-75, to further clarify the practical relevance of CdS nanoparticles in promoting bacterial growth. This addition cites existing research to explain the mechanisms of interaction between CdS and bacteria in environmental settings, highlighting the advantages of CdS in this context.
- The concentration effect of nanoparticles on bacterial growth is not clear. Maybe, at higher concentrations, an opposite effect will be observed. The author must include the concentration effect.
Response:
We have supplemented the study with data showing the effect of different concentrations of CdS nanoparticles on the growth of E. coli. In fact, we have studied the effects of different doses of CdS NPs on E. coli growth, and this has been added to the manuscript. The results indicate that a concentration of 60 mg/L of CdS shows the best growth-promoting effect, while the growth promotion at other concentrations is lower than that observed at 60 mg/L. The relevant data and figure are included in the revised supplementary material as Fig. S9. Additionally, we have added an explanation regarding the concentration effect in the manuscript, which can be found on line 356-361.
- In section 2.3, it is not clear, how CdS-E. coli was prepared.
Response:
We would like to clarify the preparation process of CdS-E. coli. Synthesized CdS was added to the E. coli bacterial culture at a concentration of 60 mg/L. The initial inoculation density of E. coli in all treatment groups was set to OD600 = 0.12. The cultures were incubated statically in M9 medium at 28°C. For the light exposure group, a full-wavelength xenon lamp (ranging from 200 to 1100 nm) was used as the light source with an intensity of 1500 lux, while the dark group was incubated under the same conditions but wrapped in aluminum foil to block the light. We have already incorporated this experimental method into lines 140-142 and 156-160 of the manuscript.
- TEM results show aggregates and make it difficult to understand the size and shape.
Response:
The primary purpose of the TEM analysis was to confirm the morphology and structure of the synthesized material, to ensure that the crystalline structure was consistent with CdS NPs, and to compare the lattice fringes with diffraction rings to validate the crystal structure of CdS. Initially, we used ImageJ to extract and analyze the spherical particles from Fig. 1 for statistical purposes. However, we acknowledge that this method might introduce some inaccuracies in size estimation.
Fig. 1 TEM analysis of CdS NPs at the 50 nm scale.
In response to your suggestion for more precise size data, we have supplemented the analysis with dynamic light scattering (DLS) to obtain a more accurate particle size distribution. As a result, we have replaced the particle size distribution graph in Fig. S1D accordingly. Additionally, we have revised the text in line 261 to reflect the DLS remeasurement of the CdS NPs size.
- Authors should elaborate on the practical relevancy of the work with logical explanations. For instance, in discussion, line 416-418, degradation of environmental pollutants with E. coli? How? You first, treat with NPs, and release the bacteria in the environment? How long does the effect remain in natural conditions? Is E. coli, appropriate bacteria to do this? If not, that choose relevant bacteria for study and than only made such claims.
Response:
To further validate the feasibility of the CdS-E. coli system in environmental pollutant degradation, we designed and conducted additional experiments aimed at addressing your questions.
Firstly, although E. coli as a common bacteria may have limitations in natural environmental applications, it serves as a reliable model to elucidate the interactions between CdS nanoparticles and microorganisms, as well as their potential in pollutant degradation. We chose the environmentally widespread bacterium E. coli and the common semiconductor mineral CdS to establish a model that could be applied to environmental pollutant remediation. This provides an experimental basis for selecting other more suitable strains for environmental applications in the future (such as bacteria that can survive in harsher conditions) or other semiconductor minerals.
To address the practical application of CdS nanoparticles combined with bacteria in environmental settings, and to solve the issues of stability and reuse, we used encapsulation technology to prepare CdS-E. coli beads. This method not only controls the release of nanoparticles but also enhances their stability and degradation capability in natural environments. Encapsulation extends the activity of both nanoparticles and bacteria, allowing them to function for a longer period under natural conditions and achieve longer-term pollutant degradation.
To this end, we conducted the following experimental steps to verify the actual effectiveness of this system in dye degradation and its potential for reuse:
- Sample Preparation: After 48 hours of light incubation, 30 mL of the CdS-E. coli sample was centrifuged at 14000×g at 4°C. The sample was washed three times with pH 7.2 PBS buffer and resuspended in 5 mL of M9 medium. The resuspension was then mixed with 2 mL of sterilized sodium alginate solution (0.1 g/mL), and the mixture was dropped into 0.1 mol/L CaCl2 solution using a syringe to prepare encapsulated beads.
- Control Group Preparation: Using the same steps, we prepared control encapsulated beads from E. coli and CdS solutions, which were used in subsequent experiments.
- Degradation Experiment: The three groups of encapsulated beads were used for decolorization experiments with 50 mg/L amaranth dye. By comparing the decolorization efficiency of different treatment groups, we determined the practical applicability of the CdS-E. coli system in dye degradation. At a wavelength of 520 nm, the decolorization effect of different encapsulated beads on amaranth dye was detected.
- Biological Safety Assessment: After the decolorization experiment, we evaluated the biological safety of the supernatant. First, 100 μL of the decolorized supernatant was added to the E. coli culture medium to observe its effects on cell growth and toxicity. Additionally, the following control groups were established:
100 μL of 50 mg/L amaranth dye solution was added to evaluate its effect on cell growth.
100 μL of PBS buffer was added as a negative control to assess E. coli growth without dye or nanoparticle treatment.
- Repeated Degradation Capacity Test: After the CdS-E. coli encapsulated bead decolorization experiment, the beads were washed three times with PBS buffer and reactivated in M9 medium at 28°C for 24 hours. The amaranth dye degradation experiment was then repeated to evaluate the beads' degradation capacity after multiple uses.
Through these experiments, we validated the potential of the CdS-E. coli system for use in environmental pollutant degradation, particularly in dye decolorization, and further explored its biological safety and reusability.
Fig. 2 (A) Dye decolorization in different treatment groups and (B) the cyclic decolorization performance of CdS-E. coli encapsulated beads on amaranth dye
From Fig. 2A, it is evident that the CdS-E. coli encapsulated beads exhibited the highest performance in the amaranth dye decolorization experiment, achieving a decolorization rate of 41.06% within 24 h. In comparison, CdS alone achieved a decolorization rate of 14.62%, while E. coli alone showed a rate of 17.66%. Notably, the amaranth dye itself did not display significant spontaneous decolorization.
Fig. 3 Biological safety assessment of CdS-E. coli encapsulated beads.
When the CdS-E. coli encapsulated beads were reused multiple times, the decolorization efficiency remained consistently around 40% for the first three cycles, indicating strong stability and the potential for repeated use in treating dye pollutants. Furthermore, we assessed the biological toxicity of the supernatant after decolorization. The results demonstrated that the treated supernatant did not exhibit significant toxicity, whereas the original amaranth dye solution significantly inhibited the growth of E. coli.
These findings suggest that CdS-E. coli encapsulated beads not only possess excellent environmental compatibility but also demonstrate strong potential for repeated dye degradation. However, after four cycles, a slight decline in decolorization efficiency was observed, indicating certain limitations in reuse. Additionally, the system's overall capacity for amaranth dye degradation still requires further optimization, which will be the focus of our subsequent research.
The above experimental results confirm the feasibility of using this system for environmental remediation. Furthermore, we have supplemented the manuscript with relevant real-world case studies supporting the practical application of this system, as outlined in the revised manuscript on lines 556-560.
We have added the section on dye degradation, reusability, and biological safety assessment to lines 237-252 and 490-501 of the manuscript.
We hope these additional experimental results address your concerns. Through these supplementary experiments, we have provided a thorough evaluation of the feasibility of this technology and presented initial scientific evidence for its potential application in real-world environmental condition.
Reference
- Lu, A.; Li, Y.; Ding, H.; Xu, X.; Li, Y.; Ren, G.; Liang, J.; Liu, Y.; Hong, H.; Chen, N.; et al. Photoelectric conversion on Earth’s surface via widespread Fe- and Mn-mineral coatings. Proceedings of the National Academy of Sciences 2019, 116, 9741-9746, doi:10.1073/pnas.1902473116.
- Wang, B.; Xiao, K.; Jiang, Z.; Wang, J.; Yu, J.C.; Wong, P.K. Biohybrid photoheterotrophic metabolism for significant enhancement of biological nitrogen fixation in pure microbial cultures. Energy & Environmental Science 2019, 12, 2185-2191, doi:10.1039/C9EE00705A.

Reviewer 2 Report
Comments and Suggestions for Authors
The effects of CdS nanoparticles on the growth and metabolisms of E.coli bacteria have been investigated. The subject is interesting and potentially publishable. However, there are many points which should be addressed, clarified and discussed by the authors in a revised version before a further consideration. Therefore, I suggest major revision of the manuscript according to the following comments:
1. The morphology of the CdS nanoparticles and their size distribution should be given. SEM and DLS can be used in this regard.
2. The potential toxicity of Cd-based nanomaterials was previously reported in (Influence of heavy nanocrystals on spermatozoa and fertility of mammals). This should be mentioned in the introduction section for further completion of the literature review.
3. Could the authors give some data relating to the surface charge of the particles? Zeta potential measurement can be used in this regard.
4. It has been mentioned that “When CdS NPs are photoexcited, they generate electron-hole pairs that have strong antibacterial effects, most of which are caused by the release of heavy metal ions and oxidative stress brought on by photoexcitation”. This statement needs to be further supported by, e.g., Vertically aligned ZnO@CdS nanorod heterostructures for visible light photoinactivation of bacteria.
5. The authors should comment on the Cd ion release capability of the sample. Then what about the stability in the antibacterial activity of the samples after long time? Also, what about the potential toxicity of the Cd ions against animals, plans and human health? These points should be addressed and clarified in the revised version.
6. The known mechanisms describing the effects of nanomaterials on the bacteria are listed by: 1) physical direct interaction of extremely sharp edges of nanomaterials with cell wall membrane (Toxicity of graphene and graphene oxide nanowalls against bacteria), 2) ROS generation (ROS generation by reduced graphene oxide (rGO) induced by visible light showing antibacterial activity: comparison with graphene oxide) even in dark (Insight into the mechanism of antibacterial activity of ZnO: surface defects mediated reactive oxygen species even in the dark), 3) trapping the bacteria within the aggregated nanomaterials (Wrapping bacteria by graphene nanosheets for isolation from environment, reactivation by sonication, and Inactivation by near-infrared irradiation), 4) oxidative stress [ACS Nano 2011, 5, 9, 6971–6980], 5) interruption in the glycolysis process of the bacteria (Escherichia coli bacteria reduce graphene oxide to bactericidal graphene in a self-limiting manner), 6) DNA damaging (Engineered ZnO and TiO2 nanoparticles induce oxidative stress and DNA damage leading to reduced viability of Escherichia coli), 7) metal ion release (Superior antibacterial activity of zinc oxide/graphene oxide composites originating from high zinc concentration localized around bacteria), and recently 8) contribution in generation/explosion of nanobubbles (Oxygen-rich graphene/ZnO2-Ag nanoframeworks with pH-switchable catalase/peroxidase activity as O2 nanobubble-self generator for bacterial inactivation). These mechanisms should be addressed in the revised version. Then, the dominant mechanism(s) occurred in this work should be discussed using suitable supports based on the mechanisms proposed in the literatures.
7. The chemical composition of the nanoparticles is not clear. FTIR can help in this regard.
8. Could the authors comment on the Na residuals in the samples? EDX can be used to check the purity of the samples.
Comments on the Quality of English Language
Minor editing of English language required.
Author Response
Dear editor:
Thank you very much for reviewing my paper and giving me so many useful comments and suggestions. I have already revised my paper accordingly, and the details are as following:
Reviewer 2:
- The morphology of the CdS nanoparticles and their size distribution should be given. SEM and DLS can be used in this regard.
Response:
Regarding the morphology and size distribution of CdS NPs, we provided a detailed supplement in the manuscript. The SEM analysis of CdS NPs has been included in Fig. S1C. Additionally, we performed dynamic light scattering (DLS) analysis to obtain the particle size distribution, and replaced the original size analysis in the manuscript with the updated Fig. S1D.
Fig. 1 TEM image of (A) CdS NPs, (B) HRTEM image of CdS NPs, (C) SEM image of CdS NPs at 2 μm and (D) the particles size distribution. (E) XRD patterns of CdS NPs.
- The potential toxicity of Cd-based nanomaterials was previously reported in (Influence of heavy nanocrystals on spermatozoa and fertility of mammals). This should be mentioned in the introduction section for further completion of the literature review.
Response:
We acknowledge the relevance of this concern, particularly given the findings from the study you mentioned, Influence of heavy nanocrystals on spermatozoa and fertility of mammals. In response, we have now included a discussion on the potential toxicity of CdS NPs in the introduction section of the revised manuscript. This addition emphasizes previous studies that have demonstrated the risks of cadmium-based materials, including their effects on mammalian fertility, as well as the broader implications for environmental and biological systems. The text regarding the added content could be found between lines 59-65 in revised manuscript.
By incorporating this literature, we aim to provide a more comprehensive review of the potential risks associated with Cd-based nanomaterials, ensuring that readers are aware of both their technological applications and the possible safety concerns. We believe this will enhance the completeness of the literature review and provide necessary context for our study’s environmental and biological relevance.
- Could the authors give some data relating to the surface charge of the particles? Zeta potential measurement can be used in this regard.
Response:
In response to your suggestion, we have added the zeta potential measurement for CdS NPs, as shown in Fig. S2, with a zeta potential value of 33.93 mV.
This zeta potential indicates that the CdS NPs exhibit moderate stability in solution, preventing excessive dispersion. This is crucial for reducing the release of free Cd²⁺ ions. Due to the surface charge of the CdS NPs, they do not readily aggregate or sediment, which helps limit the amount of Cd²⁺ ions released into the culture environment, thereby reducing their potential toxicity to bacteria. This controlled release ensures that CdS NPs can function effectively in specific environments while minimizing negative impacts on microbes and the ecosystem.
Additionally, a detailed explanation of the zeta potential characterization results has been included in the revised manuscript on line 261.
- It has been mentioned that “When CdS NPs are photoexcited, they generate electron-hole pairs that have strong antibacterial effects, most of which are caused by the release of heavy metal ions and oxidative stress brought on by photoexcitation”. This statement needs to be further supported by, e.g., Vertically aligned ZnO@CdS nanorod heterostructures for visible light photoinactivation of bacteria.
Response:
Thank you for your valuable suggestion regarding the antibacterial effects of CdS NPs caused by the release of heavy metal ions and oxidative stress induced by photoexcitation. To strengthen our discussion, we have now cited the study titled "Vertically aligned ZnO@CdS nanorod heterostructures for visible light photoinactivation of bacteria." This paper confirms that when CdS is paired with another semiconductor such as ZnO, the production of electron-hole pairs is significantly enhanced under visible light, leading to efficient bacterial inactivation through the generation of reactive oxygen species (ROS), which are the primary cause of oxidative damage to bacterial cells.
This mechanism aligns with our findings and supports the antibacterial properties of CdS NPs, which are primarily derived from their photocatalytic activity, particularly through ROS-mediated oxidative stress and the release of Cd2+ ions. We have updated the introduction section with relevant literature and enhanced the discussion on the role of photoexcited CdS NPs in microbial inactivation. The newly added explanation text can be found in lines 34-41.
- The authors should comment on the Cd ion release capability of the sample. Then what about the stability in the antibacterial activity of the samples after long time? Also, what about the potential toxicity of the Cd ions against animals, plans and human health? These points should be addressed and clarified in the revised version.
Response:
We conducted a series of experiments to systematically assess the Cd2+ release capacity of CdS NPs and their antibacterial activity stability.
Firstly, we measured the concentration of free Cd2+ in the culture medium after incubating CdS-E. coli for 48 h using ICP-MS, which showed a Cd2+ concentration of 426.78 μg/L (Fig. 2 and in Fig. S8A of revised manuscript). Based on the zeta potential measurements, we found that at an initial concentration of 60 mg/L of CdS NPs, the Cd2+ release rate was only 0.71%. This indicates that the majority of the Cd remains in the form of semiconductor CdS NPs within the system, without releasing large amounts of Cd2+. Under light conditions, these CdS NPs provide photoelectrons, further promoting the growth of E. coli, suggesting that the antibacterial effects of CdS NPs are not significantly impacted by Cd2+ release.
Fig. 2 The concentration of Cd2+ released in the culture medium from CdS-E. coli.
In response to your concerns regarding the stability of antibacterial activity, we conducted further experiments to evaluate the effects of different forms of Cd on E. coli growth. We added 426 μg/L of CdCl2 solution to the E. coli culture medium to simulate the same concentration of Cd2+ and assess its toxicity on bacterial growth. Simultaneously, we set up a treatment group with 60 mg/L of CdS NPs and a control group without any Cd addition. All three samples were incubated at 28°C in 1500 lux light intensity, and the growth curves were recorded. The results were shown in Fig. 3 or Fig. S8B of revised manuscript.
Fig. 3 The bacterial growth curves of different treatment groups.
The experimental results showed that 426 μg/L of Cd2+ significantly inhibited the growth of E. coli. However, the E. coli treated with 60 mg/L of CdS NPs exhibited the highest growth promotion effect, significantly exceeding the control group. This indicates that while the Cd2+ release was comparable to the CdCl2 group, the photoelectrons generated by CdS NPs under light conditions strongly promoted bacterial growth, surpassing the toxic effects of Cd2+. Thus, the overall growth promotion effect of the CdS group explained why the E. coli biomass accumulation was significantly higher under light conditions. We have added the above supplemental experimental details to the revised manuscript in lines 350-361, and the corresponding results can be found in the revised supplementary materials, Fig. S7.
Moreover, the detected Cd²⁺ concentration (426.78 μg/L or 0.43 mg/L) in our system remains far below the toxic dose observed in mammals, such as rodents, where acute exposure to 3 mg/kg Cd²⁺ causes severe kidney damage[1]. This concentration is significantly higher than the Cd²⁺ released in our experiment, indicating that, at the levels observed in our study, the effects on mammals would be minimal.
While these toxic effects exist, under our experimental conditions, the Cd2+ release is minimal, and the growth-promoting effect of photoelectrons far exceeds the inhibitory effect of Cd2+. Therefore, CdS NPs still demonstrate strong positive antibacterial effects. We have incorporated a discussion on the environmental toxicity effects of Cd²⁺ release dosage based on your suggestion. Please refer to lines 498-507.
To further validate the feasibility of the CdS-E. coli system in environmental pollutant degradation, we designed and conducted additional experiments aimed at addressing your questions.
Firstly, although E. coli as a model organism may have limitations in natural environmental applications, it serves as a reliable model to elucidate the interactions between CdS NPs and microorganisms, as well as their potential in pollutant degradation. This provides an experimental basis for selecting other more suitable strains for environmental applications in the future.
To address the practical application of CdS NPs combined with bacteria in environmental settings, we used encapsulation technology to prepare CdS-E. coli beads. This method not only controls the release of nanoparticles but also enhances their stability and degradation capability in natural environments. Encapsulation extends the activity of both nanoparticles and bacteria, allowing them to function for a longer period under natural conditions.
To this end, we conducted the following experimental steps to verify the actual effectiveness of this system in dye degradation and its potential for reuse:
The newly added experimental methods and results can be found in lines 237-252 and 490-501 of the revised manuscript.
Through these experiments, we validated the potential of the CdS-E. coli system for use in environmental pollutant degradation, particularly in dye decolorization, and further explored its biological safety and reusability.
Fig. 4 (A) Dye decolorization in different treatment groups and (B) the cyclic decolorization performance of CdS-E. coli encapsulated beads on amaranth dye
From Fig. 4A, it is evident that the CdS-E. coli encapsulated beads exhibited the highest performance in the amaranth dye decolorization experiment, achieving a decolorization rate of 41.06% within 24 hours. In comparison, CdS alone achieved a decolorization rate of 14.62%, while E. coli alone showed a rate of 17.66%. Notably, the amaranth dye itself did not display significant spontaneous decolorization.
Fig. 5 Biological safety assessment of CdS-E. coli encapsulated beads.
After multiple reuses, the CdS-E. coli encapsulated beads maintained a decolorization efficiency of approximately 40% over the first three cycles, demonstrating strong stability and potential for repeated use in dye pollutant treatment. The biological toxicity assessment of the supernatant showed no significant toxicity, while the original amaranth dye solution inhibited E. coli growth. These results highlight the environmental compatibility and reusability of the beads. However, a slight decline in efficiency was observed after the fourth cycle, indicating some limitations that require further optimization in future research.
The above experimental results confirm the feasibility of using this system for environmental remediation. Furthermore, we have supplemented the manuscript with relevant real-world case studies supporting the practical application of this system, as outlined in the revised manuscript on lines 556-560.
- The known mechanisms describing the effects of nanomaterials on the bacteria are listed by: 1) physical direct interaction of extremely sharp edges of nanomaterials with cell wall membrane (Toxicity of graphene and graphene oxide nanowalls against bacteria), 2) ROS generation (ROS generation by reduced graphene oxide (rGO) induced by visible light showing antibacterial activity: comparison with graphene oxide) even in dark (Insight into the mechanism of antibacterial activity of ZnO: surface defects mediated reactive oxygen species even in the dark), 3) trapping the bacteria within the aggregated nanomaterials (Wrapping bacteria by graphene nanosheets for isolation from environment, reactivation by sonication, and Inactivation by near-infrared irradiation), 4) oxidative stress [ACS Nano 2011, 5, 9, 6971–6980], 5) interruption in the glycolysis process of the bacteria (Escherichia colibacteria reduce graphene oxide to bactericidal graphene in a self-limiting manner), 6) DNA damaging (Engineered ZnO and TiO2nanoparticles induce oxidative stress and DNA damage leading to reduced viability of Escherichia coli), 7) metal ion release (Superior antibacterial activity of zinc oxide/graphene oxide composites originating from high zinc concentration localized around bacteria), and recently 8) contribution in generation/explosion of nanobubbles (Oxygen-rich graphene/ZnO2-Ag nanoframeworks with pH-switchable catalase/peroxidase activity as O2 nanobubble-self generator for bacterial inactivation). These mechanisms should be addressed in the revised version. Then, the dominant mechanism(s) occurred in this work should be discussed using suitable supports based on the mechanisms proposed in the literatures.
Response:
In response to your concerns regarding the toxicity mechanisms of CdS NPs, the study showed that the synthesized CdS NPs primarily produce ROS, which leads to toxicity. We have systematically discussed the potential mechanisms of nanomaterial effects on bacteria based on existing literature and our experimental results, as follows:
Physical contact and encapsulation mechanism: Our morphological observations and zeta potential analysis show that the synthesized CdS NPs have a spherical structure and do not exhibit sharp nanosheet features like graphene or graphene oxide. The measured zeta potential is -33.93 mV in Fig. S2, indicating that the CdS NPs carry a negative surface charge and have limited solubility in the solution. Therefore, they are unlikely to form aggregates or encapsulate E. coli, nor do they cause toxicity through direct physical contact with bacterial cells.
Fig. 6 The effects of different concentrations of CdS NPs on the enzyme activities of (A) SOD, (B) CAT, and (C) POD in E. coli.
ROS generation and antioxidant mechanism: We conducted additional experiments to measure ROS-related enzyme activity in E. coli treated with different concentrations of CdS NPs. The results showed that as the concentration of CdS NPs increased, the activity of antioxidant enzymes (such as CAT, SOD, POD) in E. coli significantly increased. This indirectly suggests that ROS generation did increase in the presence of CdS NPs. However, under light exposure, the photoelectrons generated by CdS NPs greatly enhanced the bacteria’s antioxidant capacity. As a result, the overall effect on bacterial growth was promotive, rather than inhibitory.
Metal ion release and toxicity mechanism: The experimental data indicated that the release rate of Cd2+ from CdS NPs was only 0.71%, corresponding to a Cd2+ concentration of 426.78 μg/L in Fig. 2. While Cd2+ can exhibit toxicity at high concentrations, the photoelectrons generated by CdS NPs significantly promoted glycolytic enzyme activity in the bacteria (validated by qPCR), enhancing bacterial growth and division. The qPCR analysis showed that the expression of FtsZ, a key protein involved in bacterial cell division, was not significantly affected. This suggests that DNA replication was likely not impacted during the experiment, as FtsZ is crucial for the formation of the Z-ring, a structure essential for cell division, which depends on accurate DNA replication. Therefore, the positive effects of the photoelectrons outweighed the inhibitory effects of Cd2+, resulting in an overall promotion of bacterial growth.
Nanobubble mechanism: The generation of nanobubbles typically occurs in oxygen-rich environments. Since the CdS NPs-E. coli culture environment in this study is not such an oxygen-enriched system, we believe that nanobubbles are not a significant toxicity mechanism in our research.
We have supplemented the discussion with relevant content, specifically addressing your concerns in the revised manuscript at line 499-511, and cited the references mentioned in your comments. We hope these additions adequately address your questions. If you have any further feedback, we are happy to continue refining and improving our work.
- The chemical composition of the nanoparticles is not clear. FTIR can help in this regard.
Response:
To address your concern regarding the chemical composition of the nanoparticles, we have performed Fourier Transform Infrared (FTIR) analysis, and the data has been added (Fig. S3). The absorption peaks in the range of 400–600 cm-1 correspond to the stretching vibrations of Cd-S bonds, confirming the presence of chemical bonding between cadmium and sulfur in the sample. Additionally, the peaks observed around 3000–3600 cm⁻¹ are attributed to hydroxyl (-OH) groups, likely related to water molecules adsorbed on the surface of the nanoparticles. The FTIR image could be found in the revised supplementary information as Fig. S3.
These FTIR results clearly illustrate the chemical composition of CdS NPs, further validating their structure and surface properties, addressing your concern regarding the composition of the particles. The added text for the analysis based on the FTIR spectrum detection of the sample can be found in the revised manuscript, lines 274-277.
- Could the authors comment on the Na residuals in the samples? EDX can be used to check the purity of the samples.
Response:
We have supplemented the manuscript with an EDX analysis to assess the residual Na content in the samples. The EDX results indicate that the atomic percentage of Na is 2.05%, as shown in Fig. 7.
Fig. 7 EDX analysis of the sample.
The detected Na content of 2.05% was relatively low, and based on the nature of our samples and synthesis process, we believe this did not significantly affect the overall properties of the CdS NPs. The presence of Na might be attributed to residuals from the synthesis medium, but it was within acceptable limits for the purity of the nanoparticles in this context. Thus, we concluded that Na impurities did not compromise the functionality of the samples.
Reference
- Ryabova, Y.V.; Minigalieva, I.A.; Sutunkova, M.P.; Klinova, S.V.; Tsaplina, A.K.; Valamina, I.E.; Petrunina, E.M.; Tsatsakis, A.M.; Mamoulakis, C.; Stylianou, K.; et al. Toxic Kidney Damage in Rats Following Subchronic Intraperitoneal Exposure to Element Oxide Nanoparticles. Toxics 2023, 11, doi:10.3390/toxics11090791.

Reviewer 3 Report
Comments and Suggestions for Authors
The study by Yang et. al is well-designed, with clear objectives focusing on the effects of CdS NPs on E. coli under varying light conditions. The methods for evaluating enzyme activities, gene expression, and electrochemical properties are appropriate and well-described. However, there are some concerns which have been described below.
1. Section 2.3: The description lacks details on the conditions under which the CdS-E. coli and E. coli cultures were incubated (e.g., temperature, medium used, initial inoculum density). These factors are crucial for understanding the growth dynamics and interactions with CdS, and they should be specified to ensure reproducibility.
2. Section 3.2: The reported changes in NADP-MDH and NADP-IDH activities under illumination are intriguing, but the underlying mechanisms are not fully explored. The significant decrease in NADP-IDH activity, in particular, warrants further investigation to understand its implications for carbon metabolism.
3. The study reports changes in enzyme activities but does not discuss whether these changes are due to alterations in enzyme kinetics (e.g., Vmax, Km). Understanding how the kinetics are affected would provide deeper insights into the metabolic impact of CdS NPs.
4. The discussion section should be expanded to consider potential limitations, alternative explanations for the results, and the broader implications of the findings.
Author Response
Dear editor:
Thank you very much for reviewing my paper and giving me so many useful comments and suggestions. I have already revised my paper accordingly, and the details are as following:
Reviewer 3:
- Section 2.3:The description lacks details on the conditions under which the CdS-E. coli and E. coli cultures were incubated (e.g., temperature, medium used, initial inoculum density). These factors are crucial for understanding the growth dynamics and interactions with CdS, and they should be specified to ensure reproducibility.
Response:
Thank you for your question. We would like to clarify the preparation process of CdS-E. coli. Synthesized CdS was added to the E. coli bacterial culture at a concentration of 60 mg/L. The initial inoculation density of E. coli in all treatment groups was set to OD600 = 0.12. The cultures were incubated statically in M9 medium at 28°C. For the light exposure group, a full-wavelength xenon lamp (ranging from 200 to 1100 nm) was used as the light source with an intensity of 1500 lux, while the dark group was incubated under the same conditions but wrapped in aluminum foil to block the light. We have already incorporated this experimental method into lines 140-142 and 156-161 of the manuscript.
- Section 3.2:The reported changes in NADP-MDH and NADP-IDH activities under illumination are intriguing, but the underlying mechanisms are not fully explored. The significant decrease in NADP-IDH activity, in particular, warrants further investigation to understand its implications for carbon metabolism.
Response:
After carefully considering the initial results, we realized that the first NADP-IDH measurement was conducted using stored samples, which may have affected the enzyme activity. To address this issue, we repeated the NADP-IDH measurement using fresh samples. The new measurements clearly showed that, under light exposure, CdS NPs significantly increased the NADP-IDH activity within E. coli, consistent with the observed increase in the activity of other carbon metabolism enzymes under photoelectron stimulation.
Additionally, we measured fructose-1,6-bisphosphate aldolase activity to further verify the promoting effect of photoelectrons on carbon metabolism enzyme activity. The increase in aldolase activity within E. coli under illumination indicates an enhanced ability to drive carbon cycling, as this enzyme plays a critical role in the glycolysis and gluconeogenesis pathways by catalyzing the cleavage of fructose-1,6-bisphosphate into triose phosphates. This improved carbon cycling supports faster growth and higher metabolic efficiency in E. coli.
Fig.1 Differences in the activity units of Fructose-1,6-bisphosphate aldolase in carbon metabolism by different treatment groups.
Both enzymes showed a consistent increase in activity under illumination, indicating that photoelectrons indeed enhance the activity of key carbon metabolism enzymes in the bacterial system. This supports our conclusion that light-induced electron transfer plays a significant role in promoting bacterial carbon metabolism.
We have updated the manuscript with these new data and revised the text regarding NADP-IDH enzyme activity, as reflected in lines 393-398 of the revised manuscript, to capture the corrected results and their implications for understanding how photoelectron stimulation affects bacterial enzyme activity. Future studies will further explore the specific mechanisms by which photoelectrons influence the carbon metabolism enzyme activity in E. coli.
- The study reports changes in enzyme activities but does not discuss whether these changes are due to alterations in enzyme kinetics (e.g., Vmax, Km). Understanding how the kinetics are affected would provide deeper insights into the metabolic impact of CdS NPs.
Response:
We understand the importance of exploring whether the observed changes in enzyme activities are linked to alterations in enzyme kinetics, such as Vmax and Km. To address this, we conducted Michaelis-Menten kinetic studies on two key enzymes: POD as an antioxidant enzyme, and NADH-IDH as a carbon metabolism enzyme.
In this study, enzyme activities were measured using enzyme activity assay kits. we are happy to address your concerns and conducted enzyme kinetic measurements accordingly in Fig. 2.
Fig. 2 (A) NADP-IDH and (B) POD enzyme activity changes in E. coli and CdS NPs-E. coli.
By fitting the Michaelis-Menten enzyme kinetic curves, we found that the addition of CdS NPs did not alter the intrinsic kinetic parameters of E. coli enzymes (Vmax and Km). The observed increase in enzyme activity was not due to changes in these kinetic parameters but was instead the result of CdS NPs acting as external electron donors, promoting the bacterial metabolic processes. This enhancement in bacterial metabolism positively regulated both antioxidant and carbon metabolism enzyme activities, leading to significant increases in their activities.
Fig. 3 Effects of different concentrations of CdS NPs on the enzyme activities of (A) NADP-IDH and (B) POD in E. coli.
Building on the kinetic studies of the two enzymes, we further investigated the effects of different concentrations of CdS NPs on the activities of these enzymes in E. coli. The results showed that 60 mg/L CdS NPs had the most pronounced effect in enhancing enzyme activity. This finding is consistent with the differences observed in the E. coli growth curves at different concentrations, further confirming our earlier conclusion that CdS NPs, as external electron donors, boost carbon metabolism and antioxidant capacity in E. coli.
Fig. 4 Effect of different concentrations of CdS on E. coli growth.
We have incorporated these kinetic data into the manuscript, which further supports the positive role of photoelectrons in enhancing enzyme activities within the bacterial system. We believe these results address your concerns well. Future work will compare the kinetic effects under conditions without photoelectron stimulation to provide a more comprehensive understanding of CdS NPs-induced metabolic changes.
- The discussion section should be expanded to consider potential limitations, alternative explanations for the results, and the broader implications of the findings.
Response:
We acknowledge your concerns regarding the practical application of our research. To assess the feasibility of the CdS-E. coli system in environmental pollutant degradation, we conducted additional experiments. While E. coli may have limitations as a model for natural environments, it effectively demonstrates the interactions between CdS NPs and bacteria. This forms a foundation for selecting more suitable strains for future environmental applications.
To enhance practical applicability, we utilized encapsulation technology to stabilize CdS-E. coli beads, controlling nanoparticle release and improving degradation capacity. This method extends the functional lifespan of both CdS NPs and bacteria in environmental conditions.
The newly added experimental methods and results can be found in lines 237-252 and 490-501 of the revised manuscript. These experiments validate the potential of the CdS-E. coli system for dye degradation and explore its biological safety and reusability.
Fig. 5 (A) Dye decolorization in different treatment groups and (B) the cyclic decolorization performance of CdS-E. coli encapsulated beads on amaranth dye
From Fig. 1A, it is evident that the CdS-E. coli encapsulated beads exhibited the highest performance in the amaranth dye decolorization experiment, achieving a decolorization rate of 41.06% within 24 h. In comparison, CdS alone achieved a decolorization rate of 14.62%, while E. coli alone showed a rate of 17.66%. Notably, the amaranth dye itself did not display significant spontaneous decolorization.
Fig. 6 Biological safety assessment of CdS-E. coli encapsulated beads.
CdS-E. coli encapsulated beads maintained around 40% decolorization efficiency for the first three cycles, showing good stability and potential for repeated dye treatment. After four cycles, a slight decline in efficiency was observed. The treated supernatant showed no significant toxicity, unlike the original amaranth dye solution, which inhibited E. coli growth. These results suggest excellent environmental compatibility and potential for repeated dye degradation, though further optimization is needed.
We confirmed the feasibility of this system for environmental remediation and provided real-world case studies in the revised manuscript (lines 471-476).
We also assessed Cd2+ release, which was minimal at 426.78 μg/L, with a release rate of 0.71%. This indicates that most Cd remained in its semiconductor form, and under light, the generated photoelectrons promoted bacterial growth, minimizing the impact of Cd2+ release on antibacterial activity.
Fig. 7 The concentration of Cd2+ released in the culture medium from CdS-E. coli.
In response to your concerns regarding the stability of antibacterial activity, we conducted further experiments to evaluate the effects of different forms of Cd on E. coli growth. We added 426 μg/L of CdCl2 solution to the E. coli culture medium to simulate the same concentration of Cd2+ and assess its toxicity on bacterial growth. Simultaneously, we set up a treatment group with 60 mg/L of CdS NPs and a control group without any Cd addition. All three samples were incubated at 28°C in 1500 lux light intensity, and the growth curves were recorded. The results were shown in Fig. 8 or Fig. S8B of revised manuscript.
Fig. 8 The bacterial growth curves of different treatment groups.
The experimental results showed that 426 μg/L of Cd2+ significantly inhibited the growth of E. coli. However, the E. coli treated with 60 mg/L of CdS NPs exhibited the highest growth promotion effect, significantly exceeding the control group. This indicates that while the Cd2+ release was comparable to the CdCl2 group, the photoelectrons generated by CdS NPs under light conditions strongly promoted bacterial growth, surpassing the toxic effects of Cd2+. Thus, the overall growth promotion effect of the CdS group explained why the E. coli biomass accumulation was significantly higher under light conditions. We have added the above supplemental experimental details to the revised manuscript in lines 324-329, and the corresponding results can be found in the revised supplementary materials, Fig. S7.
Moreover, the detected Cd2+ concentration (426.78 μg/L or 0.43 mg/L) in our system remains far below the toxic dose observed in mammals, such as rodents, where acute exposure to 3 mg/kg Cd²⁺ causes severe kidney damage[1]. This concentration is significantly higher than the Cd²⁺ released in our experiment, indicating that, at the levels observed in our study, the effects on mammals would be minimal.
While these toxic effects exist, under our experimental conditions, the Cd2+ release is minimal, and the growth-promoting effect of photoelectrons far exceeds the inhibitory effect of Cd2+. Therefore, CdS NPs still demonstrate strong positive antibacterial effects. We have incorporated a discussion on the environmental toxicity effects of Cd²⁺ release dosage based on your suggestion. Please refer to lines 498-507.
Reference
- Ryabova, Y.V.; Minigalieva, I.A.; Sutunkova, M.P.; Klinova, S.V.; Tsaplina, A.K.; Valamina, I.E.; Petrunina, E.M.; Tsatsakis, A.M.; Mamoulakis, C.; Stylianou, K.; et al. Toxic Kidney Damage in Rats Following Subchronic Intraperitoneal Exposure to Element Oxide Nanoparticles. Toxics 2023, 11, doi:10.3390/toxics11090791.

Round 2
Reviewer 1 Report
Comments and Suggestions for Authors
The authors successfully addressed all the comments.
Comments on the Quality of English LanguageThe English language is understandable with good flow.
Reviewer 2 Report
Comments and Suggestions for Authors
The revisions are acceptable.